# Experiences of adult children caring for parents aging in place

Oluwakemi Opanubi[1]*, Jochebed Ade-Oshifogun[2]

1 Department of Nursing, Washington Adventist University, Takoma Park, Maryland, United States of America, 2 School of Nursing, New Mexico State University, Las Cruces, New Mexico, United States of America

☯ These authors contributed equally to this work.
* oopanubi@wau.edu

## Abstract

The rapid rise of the older adult population has increased the need for family caregiving. Adult children caring for their aging parents are exposed to a range of stressors, and their experiences shape care outcomes. We aim to describe these experiences using a grounded theory approach and develop a theory based on our findings. The inclusion criteria were adult children of any age who had provided care for at least one year to one or both parents, including parents-in-law, in a home setting. Participants were recruited using snowball sampling from community centers for older adults and churches. Espousing the grounded theory approach that often relies on interviews as a primary data collection method, 22 semi-structured personal interviews were conducted. Thematic analysis was completed using the NVivo software. The challenge-acceptance theory was developed through open coding, axial coding, selective coding, and continuous analysis of the data. The care recipients were aged 65 years and above. The caregivers' experiences were summarized as a complex and dynamic process with negative and positive aspects, ranging from an increased level of stress and anxiety due to uncertainties of caregiving to the profound meaning and fulfillment experienced during the caregiving process. The results of the coding process consisted of four themes: the unknown, the need for effective communication, aging parents' behavior, and caregiving. The resulting grounded theory, the challenge-acceptance theory, explains how caregivers transition from caregiving challenges to accepting the challenges. Caregivers struggle with caring for aging parents and experience severe stress, which may affect their health. Healthcare professionals should proactively identify and address these issues before significantly impacting caregivers' health. Adequate communication with caregivers, provision of helpful resources, and promotion of proactive caregiving strategies can reduce caregiver burden and stress.

**Data availability statement:** Data cannot be shared publicly because of confidentiality of data provided as approved by the IRB. Data are available from the Catholic University of America's Data Access. Contact via cua-ops@cua.edu for researchers who meet the criteria for access to confidential data.

**Funding:** The author(s) received no funding for this work.

**Competing interests:** The authors have declared that no competing interests exist.

## Introduction

Aging in place, a concept gaining increasing attention in gerontology refers to aging people living in their own homes or their adult children's homes instead of in institutional care settings [1]. This significant concept highlights the importance of maintaining independence and a sense of familiarity for aging individuals. The caregiving responsibilities typically undertaken by adult children vary according to the level of care required by their aging parents and their degrees of dependency [2,3]. Caregiving responsibilities in this context range from supporting activities of daily living to developing skills involving specialized equipment and navigating complex healthcare and social systems [3].

Caregivers face multifaceted challenges, including mental health burdens, financial strain, and a lack of support services [4]. Other challenges include dependents' worsening health and manipulative actions and the need to balance work and care commitments [2]. Three motivational themes for continuing to provide care despite the identified challenges include the intrinsic worth of caring for family, understanding caregiving itself, and making the best of the experience [2].

Studies have explored challenges specific to cultures and races. For example, culture helps explain caregivers' support needs within African American and Chinese communities [5,6]. In many cultures, traditional gender roles assign caregiving responsibilities primarily to women [5,6], who constitute most of the caregiving population [7]. In 2020, more than one in five people in the US were identified as caregivers [8], and more women were expected to take time off work for caregiving [9]. Caregivers frequently encounter substantial financial and emotional burdens, exacerbating gender disparities and placing a disproportionate burden on females [7], limiting their economic opportunities and increasing their vulnerability to physical and emotional stress [4].

Understanding caregivers' experiences is vital for implementing interventions to reduce their challenges. These experiences can be positive or negative. Positive caregiving experiences include the potential for growth, increased self-efficacy and competence, personal satisfaction, and improved relationships with the caregiving recipients [10,11]. Negative experiences include constraints on caregivers' day-to-day routines, downtime, and interpersonal connections, which can result in physical ailments and mental health issues such as depression and anxiety [3].

Grounded theory is a process whereby qualitative researchers analyze data collected from interviews to develop theories that provide a deeper understanding of phenomena. Grounded theory has been used to expound caregivers' experiences. The theory of adaptive caregiving by Mroz et al. focused on caregiving for parents with dementia [12]. This adaptive caregiving theory depicts a cyclical process caregivers use to acknowledge the loss, seeking balance before embracing a caregiving identity. In their "adaptive resourcefulness in rural caregiving" theory, Michaels and Meeker described growing into caregiving by utilizing strategies for support and resources for caregivers of older adults living at home [13]. Davis and Lee, in their grounded theory of the resilience-building process, described the resilience and burnout in family caregivers, showing that caregivers who used mindfulness and sought professional support developed resilience [14].

Conclusions from the studies reviewed suggest the need for interventions focused on improving access to healthcare services; providing timely information, education, and training; establishing support networks; implementing respite services; demonstrating self-care; and enhancing transportation services [12–14]. Interventions should be culturally appropriate, considering the significant effect of culture on caregiving [3,15].

With aging in place generating growing interest yet remaining underexplored [16], especially in a multicultural context, we aim to explore this phenomenon from the perspective of adult children as caregivers.

## Materials and methods

### Design and approach

We used a qualitative research method based on grounded theory to answer the research question, "What are adult children's experiences of providing care for aging parents?" We examined the approach to involvement in parents' care and the effects from the vantage point of the caregivers.

Strauss and Corbin's grounded theory approach was chosen for its linear method involving open, axial, and thematic coding [17,18]. This approach, which combines sociology and nursing theories, uses symbolic interactionism through interviews. We used this to understand the meanings caregivers assign to their approach to caring for their parents [19,20]. We used grounded theory as the theoretical framework because it is suitable for exploring adult children's experiences caring for parents and examining social processes and human interaction [21]. We conducted in-depth interviews to understand caregiving roles, responsibilities, and strategies to mitigate caregiver stress and burden.

### Population and sample

Adult children caring for one or both parents, including parents-in-law, who are aging in place, formed the study's population. The inclusion criteria were male and female adult children caring for aging parents in their homes for at least one year. We excluded caregiving in residential care institutions from the study. The inclusion criteria for care recipients were adults aged 65 and older who required home care.

Although snowball sampling was used, we implemented an intentional strategy to maximize diversity in the sample. Initial recruitment took place through community-based organizations, including churches and senior centers, known to serve diverse racial and ethnic populations. Initial participants were encouraged to refer caregivers from different backgrounds, including men and individuals of varied ethnicities. This approach resulted in a heterogeneous sample that included caregivers of both genders and multiple racial/ethnic groups in the Mid-Atlantic US, which enriched the data and enhanced the credibility of grounded theory development.

### Ethical considerations

This study was approved by the Committee for the Protection of Human Subjects of the Catholic University of America (Protocol No. 18–0099). After thoroughly explaining the study and addressing their questions and concerns, the participants provided written informed consent that included permission to record the interview. Owing to the possibility of emotional stress related to participants' reminiscence over challenges and difficulties experienced, professional counselors were available during and after data collection. The participants chose a convenient interview location, including their place of residence, an office, a library, or a church. We ensured confidentiality by using pseudonyms.

### Instruments and measurement

Our interview guide elicited responses regarding caregivers' experiences providing care to their parents. We used exploratory open-ended questions to introduce the semi-structured interview. The initial question — "What are your experiences of providing care for your aging parents?" — stimulated other questions based on the participants' responses. The

follow-up questions included: (a) How do you, as a caregiver, handle challenges?" and (b) "What coping skills make caregiving manageable for you?" During interviews, spoken and unspoken cues guided pertinent in-depth questioning [22].

### Member checking and credibility

An expert in grounded theory reviewed the demographic forms and interview process and guide. A pilot study with some participants ensured the clarity of the research questions. There was no need to change the questionnaire after the pilot; therefore, data from the initial participants were included in the final analysis. For member checking, the participants reviewed the transcripts and derived codes for accuracy [21]. We ensured credibility through prolonged contact with the data and robust narratives. Data quality was confirmed using an audit trail and journaling. Documentation through an audit trail ensures compliance, integrity, and accountability through data tracking.

### Researcher reflexivity and rigor

To enhance the credibility and trustworthiness of the study, we employed several strategies to reduce potential researcher bias. The lead researcher maintained a reflexive journal throughout the data collection and analysis phases. This journal was used to record assumptions, emotional responses, and evolving interpretations, which helped maintain awareness of potential subjectivity.

To minimize interpretive bias, bracketing was used to consciously set aside prior beliefs and personal experiences related to caregiving during interviews and data analysis. In addition, peer debriefing was conducted throughout the coding and theory development process with a qualitative research consultant familiar with grounded theory methodology. These sessions provided external perspectives, challenged emerging interpretations, and enhanced analytical rigor. These steps contributed to the confirmability and credibility of the Challenge-Acceptance Theory (CAT) and ensured that the findings were grounded in participants' experiences rather than the researcher's expectations.

### Data collection

Each interview took approximately 60 minutes to complete. The principal investigator used probing questions and listened effectively during the interviews, taking notes to facilitate the identification of concepts. Although 25 participants volunteered for the interview, data saturation was reached after interviewing 22 participants, causing redundancy and diminishing returns on information; hence, no further interviews were conducted. The interview commenced on March 10, 2019, and concluded on June 1, 2019.

### Data management and analysis

The principal investigator retrieved the qualitative data from memos, field notes, and interviews to document deeds and activities. The recorded interview data were transcribed verbatim for analysis. We compared and analyzed participants' data to examine evolving patterns and themes using the NVivo 12 Plus (QSR International) software for the coding process. The initial analysis involved open coding, where the principal investigator entered an exploratory field with an unbiased perspective, creating codes from the transcripts. Next, axial coding organized the patterns established during open coding. We studied the connections and relationships among patterns during axial coding. The relationships were further organized using a constant comparative method to determine connections in the data obtained by developing themes emerging from the data. With no new insights emerging, we agreed on data saturation. The final step, selective coding, ascertained the key classifications.

## Results

### Participants demographics

Table 1 presents the demographic of 22 participants. Four ethnicities were included in the sample, with more Caucasian participants (36%) than other ethnicities. Most participants (59%) were employed full-time and held a bachelor's degree (36%). There were no observable cultural differences in the participants' responses.

**Table 1. Participants' demographics.**

| Caregiver Characteristics | Category | N = 22 |
|---|---|---|
| **Gender** | Man | 9 (41%) |
| | Woman | 13 (59%) |
| **Age (Years)** | 29–55 | 9 (41%) |
| | 56–64 | 9 (41%) |
| | ≥ 65 | 4 (18%) |
| **Education** | High school | 1 (5%) |
| | Attended college, not graduated | 2 (9%) |
| | College | 1 (5%) |
| | Bachelor's | 8 (36%) |
| | Master's | 7 (32%) |
| | PhD | 3 (14%) |
| **Employment** | Full-time | 13 (59%) |
| | Part-time | 4 (18%) |
| | Retired | 5 (23%) |
| **Race/Ethnicity** | African American | 6 (27%) |
| | Asian | 3 (14%) |
| | Caucasian | 8 (36%) |
| | Hispanic | 5 (23%) |

### Themes

The main themes identified from the qualitative data analysis were the unknown (uncertainties of caregiving), the need for effective communication (communication gaps), aging parents' behavior, and caregiving. For confidentiality, participant numbers were used instead of names while presenting the participants' quotes. Table 2 presents the main themes and subthemes.

### The unknowns (uncertainties of caregiving)

One of the major themes was the "the unknown." The participants described the unknown in six different ways: inherent uncertainties (facing the unknown), limited or no caregiving support services or caregiving tools (resources), difficulty in comprehending medical aspects (understanding health conditions), the natural response of fear due to unpredictable future events (fear of the unknown), challenges mediating and coordinating care in the face of the unknown (navigating the unknown), and the subsequent impact of these challenges on the participants (effects of the unknown).

### Facing the unknown

Participants expounded on the myriad uncertainties they encountered throughout their caregiving endeavors. To the participants, facing the unknown meant navigating uncharted territories or confronting situations they least expected. The diverse array of unknown factors they experienced encompassed crucial aspects such as difficulty in identifying reliable resources, understanding their loved one's health conditions, and fear of the unknown regarding their loved one's condition and health outcomes. For example, in the initial caregiving stage, one participant believed that their aging parents had sorted out palliative and end-of-life care plans. Unbeknownst to the participant, these plans did not exist. Awareness of this situation prompted the participant to hire an attorney to assist. The participants described "facing the unknown" regarding resources, understanding health conditions, and fear of the unknown, including vulnerabilities.

**Table 2. Main themes and subthemes identified by the participants.**

| Main Themes | Number of Participants Identifying the Themes (%) | Subthemes | Number of Participants Identifying the Subthemes (%) |
|---|---|---|---|
| The Unknown (Uncertainties of Caregiving) | 22 (100%) | | |
| | | Facing the Unknown | 22 (100%) |
| | | Resources | 22 (100%) |
| | | Understanding Health Conditions | 16 (73%) |
| | | Fear of the Unknown | 20 (91%) |
| | | Navigating the Unknown | 19 (86%) |
| | | Effect of the Unknown | 22 (100%) |
| Need for Effective Communication (Communication Gaps) | 21 (96%) | | |
| Aging Parents' Behavior | 20 (91%) | | |
| Caregiving | 21 (96%) | Purposeful Caregiving (Caregiving with Purpose) | 19 (86%) |
| | | Art of Caregiving (Caregiving through "Art") | 19 (86%) |
| | | Heart of Caregiving (Caregiving from the Heart) | 19 (86%) |
| | | Meaningful Caregiving (Caregiving with Meaning) | 21 (96%) |

## Resources

Participants experiencing uncertainty regarding the initiation of their caregiving endeavors undertook a deliberate process of information acquisition, engaging in online research via platforms such as Google and WebMD to access insights into medical and care-related queries. Additionally, participants highlighted that resources derived from senior community centers, where they received valuable insights, and the transmission of knowledge through informal channels such as word-of-mouth were pivotal in enhancing their caregiving knowledge. Participants underscored the significance of helpful resources, as illustrated by the following statement:

> I would like to see or maybe be part of a group of caregivers where we share experiences and ideas. I have not found one yet, but that would be helpful. (P9)

## Understanding health conditions

Some participants were keenly interested in understanding their parents' illnesses or health conditions. They desired to cultivate an enlightened perspective on their parents' health situations, fostering awareness that would empower them with the knowledge and skills necessary to navigate and effectively respond to the complexities of their parents' health challenges.

> I wish I knew more about diabetes because my mom has that. She has low blood pressure, and I do not know what that means. I know she is tired and lethargic. (P17)

## Fear of the unknown

Participants expressed fear of the unknown regarding uncertainties about caring for older adults, illness outcomes, and vulnerability. Specific discourse centered on participants' articulation of fears and concerns about the vulnerability of the older adult demographic, reflecting a thematic exploration of the anxieties and apprehensions embedded in their caregiving narratives. One participant said:

> *Older people are like real targets. Even people he knows come by and take advantage of him financially, so I have had to put things in place that he does not know about. Somebody can just come over and ask him for $10,000, and he writes a check. It is awful.* (P2)

## Navigating the unknown

Participants expressed fear of the unknown and challenges with navigating the unknown in the caregiving experience. Caregivers often encounter situations, challenges, or health conditions that are unpredictable, unfamiliar or lack clear solutions. Navigating these unknowns involves adapting to unexpected circumstances, making informed decisions despite uncertainties, and seeking solutions to challenges that may not have straightforward answers.

Participants employed descriptive phrases such as "trying to survive," "juggling things," and "trying to figure things out" to elucidate their strategies for fulfilling the caregiving role. Within the framework of managing caregiving responsibilities, individuals leveraged familial support systems, engaging the assistance of spouses, siblings, and children. Additionally, participants availed the help of external support structures, including support groups, senior community centers, religious institutions, and friendships, as valuable resources for navigating their caregiving roles. One participant described navigating this role as follows:

> *It is a lot of that, managing; I am trying to figure things out like the groceries; the pharmacy is hard because they have different needs. Then, there are some tasks, such as dealing with doctor visits or even just talking to the doctors over the phone or their nursing staff. I have a running list of the doctors' names, addresses, phone numbers, and the nursing assistants. Then I try to have little notes for their daughters in college or check about personal things that can help me remember, more so when I'm not there.* (P17)

## Effects of the unknown

Confronted with the exigencies of caregiving responsibilities, participants unequivocally articulated the impact of the uncertainties inherent in the caregiving process. To navigate them, participants embraced structure and routine to facilitate the management of daily tasks while balancing life demands. However, even with such routines, caregivers encountered persistent challenges that exerted discernible effects on their physical, emotional/psychological, and spiritual well-being.

The physical repercussions of caregiving included fatigue, exhaustion, weight gain, reduced exercise, aches and pains, and sleep deprivation. Some participants articulated the toll these physical effects exacted on their well-being, with some experiencing impediments in their capacity to provide care.

> *The distress has caused me to have a lot of aches and pains and lack of sleep, and I get drowsy, so it has affected that.* (P11)

> *You are exhausted because you have not exercised, and knowing if I just exercise, I would have more energy and get some of the stress and need to exercise, and yet you don't see the time of the day for doing that…* (P13)

Conversely, a subset of participants reported an enhancement in health behaviors owing to the caregiving experience, exemplified by increased engagement in activities such as walking, thus highlighting a positive outcome derived from their caregiving journey.

As elucidated by participants, the emotional and psychological ramifications of caregiving encompass a spectrum of sentiments, including feeling overwhelmed, drained, frustrated, anxious, stressed/distressed, trapped, angry, burdened, guilty, and worried. Notably, some participants reported experiencing amusement by deliberately incorporating humor into their caregiving journeys.

*Well, I mean, it is depressing. There is no getting around it.* (P2)

*When I see my mother in distress, that stresses me a lot, and that cannot help but have an impact on my well-being…* (P14)

Participants detailed the impact of integrating spirituality into their care management practices. Spirituality encompassed various activities, such as prayer, meditation, music engagement, reliance on religious texts, attending church services/retreats, and an appreciation for poems. The narratives below underscore the diverse approaches employed by participants to leverage spirituality as a constructive component of their caregiving strategies.

*On the other hand, you may draw strength from your relationship with the creator because that person sustains you. There are times when I feel like I do not have enough time, and there are other times when you are calling on Him to give you the strength you need to do what you need to do on a daily basis.* (P1)

### Need for effective communication (communication gaps)

Participants elucidated their encounters with healthcare professionals who offered less than adequate communication about their aging parents. They emphasized the imperative for doctors and nurses to communicate effectively with them. The significance of effective communication, both among healthcare professionals and between participants and healthcare providers, has emerged as a crucial factor contributing to a more comprehensive understanding of the health status of their aging parents. One participant stated:

*Well, you cannot be there all day to wait for the doctor who may or may not come by and to find out the story firsthand and the channels of communication that are the really frustrating part and keeping people in the loop… Having some way of communicating so that everyone knows what is going on would be incredibly helpful.* (P13)

### Aging parents' behavior

Behavior involves navigating challenges associated with managing factors such as aging parents' stubbornness, conflicts in personalities and attitudes, and changes in behavior related to the dynamic shifts inherent in the parent-child relationship. Predominantly, participants discussed the behavioral changes in parents diagnosed with dementia or when aging parents lived with them. Participant 3 detailed the dementia-related behaviors exhibited by her father, citing instances of unpredictable irritability as notable manifestations.

*I find it incredibly challenging that my father is now unpredictably cranky and irritable with caregivers. It is challenging when it is with me, but it stresses me out beyond anything when I get these calls…* (P3)

*It is just that my mum is very strong-willed, and I am also, so it has to be her way. That is the most difficult thing.* (P4)

### Caregiving

**Purposeful caregiving (caregiving with purpose).** Participants described the experience of purposeful caregiving as involving the dual objectives of allocating time for self-care and providing care to others, focusing on enhancing aging parents' quality of life. The intentional approach involves tapping into the preferences and passions of aging parents, which may include activities such as music, dancing, and poetry. The notion of leading a purposeful life, as articulated by some participants, extends to both dedicating time to others and prioritizing personal well-being. A participant underscored the dual significance of self-care, emphasizing its role in personal well-being and as a requisite for effective caregiving.

*You have to also take care of yourself not only for your sake but in order to be a good caregiver.* (P2)

### Art of caregiving (caregiving through "art")

Participants delved into the intricacies of the "art" of caregiving, elucidating how various forms of artistic expression serve as instrumental tools for themselves and their aging parents and understanding the unique needs and preferences of the aging parent. This "art" involves attuning to the emotional well-being of parents aging in place, fostering a compassionate and supportive environment, and adapting caregiving approaches to specific situations. Caregiving requires creativity and art to enhance the quality of life of aging parents. This holistic and personalized approach acknowledges their individuality. Participants emphasized the desire to prevent idleness and the need for stimulating activities to enhance their aging parents' overall well-being and engagement.

*She is doing artwork. She's interacting with the artists that come and play music, and she really responds to that, and in the minute, in that very minute, the world is meaningful...* (P6)

Music emerged as a therapeutic resource for participants and their aging parents, providing emotional benefits. One participant, for instance, drew on knowledge acquired from YouTube videos detailing the neurological effects of music, tailoring her caregiving approach for her mother, who exhibited positive responses to music. One participant's intentional incorporation of music and dancing into caregiving practices for her mother with dementia exemplified the deliberate integration of artistic elements. Participants also claimed artistic events at the senior community center, featuring musicians, dancers, painters, opera singers, and fiddlers, providing opportunities for aging parents to partake in such activities alongside family members, which enriched caregiving experiences for the aging parents and caregivers. Moreover, participants found inspiration and encouragement in poems and Biblical quotes such as "Do not grow weary in doing what is good" and "Honor your father and mother."

### Heart of caregiving (caregiving from the heart)

The "heart" of caregiving, as it relates to purposeful caring, delves into the emotional core of caregiving, in which participants articulate vital elements such as love, understanding, patience, connection, sacrifice, forgiveness, and the preservation of dignity and respect as integral components of the caregiving experience. Against this backdrop, a prevailing and unifying motivation emerged—love. Love was described as a driving force, serving as the common purpose that impelled participants to shoulder the responsibility of caring for their loved ones. Two participants stated the following.

*It also motivates me to give an example of what true self-sacrificing love and care.* (P11)

*He is my dad, and we love them, and he did that for us. This is our honor to do it for him. It is difficult, but as I said, there is no one else to do it, and we have to do it, so that is out of honor and out of respect and love because he needs us.* (P10)

Not all participants explicitly articulated love as the primary motivation for their caregiving endeavors. One participant offered a different perspective because she had experienced a less-than-ideal upbringing. However, she believed that caring for her father was the right thing to do.

**Meaningful caregiving (Caregiving with meaning)**

Despite encountering challenges in caregiving, participants derived profound meaning from their experiences. They expressed that caregiving was a blessing and a burden; however, they perceived significance in their caregiving journey. Participants reported engaging in meaningful caregiving practices that benefited themselves and their aging parents. They described meaningful caregiving as transcending mere tasks. It involves recognizing the holistic personhood of their aging parents with its potential impact on longevity. Participants underscored how they had constructed a caregiving model for their children. Additionally, they expressed gratitude for the gift of caregiving and emphasized the importance of accepting the caregiving process.

*Sometimes, seeing an example of it versus hearing it impacts you greatly and much better, and so I think that is another thing that motivates me since I have children that I am also caring for that we can show them what that looks like, and so they have a living example.* (P11)

*There is no doubt because the level of care she is getting is keeping her alive well beyond the normal, and so there is no surrendering this task.* (P6)

## Discussion

Our findings revealed the intricate and multifaceted nature of caregiving for aging parents, particularly when faced with the unknown. Caregivers traverse significant uncertainties —whether related to understanding health conditions, accessing appropriate resources, or managing the emotional toll caregiving takes on their well-being. Many participants shared experiences of emotional, spiritual, and physical depletion, amplified by inconsistent communication with healthcare providers and the unpredictable behaviors of their aging parents, particularly those with dementia. Nevertheless, the caregivers consistently found purpose and meaning in their roles, often drawing on personal values such as love, respect, and a sense of duty. Our study demonstrates that caregiving is not simply a set of tasks but an emotional journey that caregivers undertake to enhance their loved one's quality of life while shaping their own.

The following discussion explains and compares our findings with those of similar studies. Our study had predominantly female participants. The higher number of females reveals the gender disparity in caregiving responsibilities, with women typically assuming a more prominent role than men in caring for aging parents [23]. This expectation is deeply rooted in the cultural and social norms that dictate women's roles as caregivers and homemakers, while men often fulfill provider roles outside the home [24,25]. The higher educational status in our demographics may be because the initial contact in the snowball recruitment process was well-educated. By including four ethnicities (Black, Asian, Hispanic, and Caucasian), we increased the pertinence of our findings to our demographics. Consequently, the outcomes are not specific to a single ethnic group. Although there are cultural effects to caregiving, we did not observe any variance in responses based on ethnicity or race.

Compared with other studies, we found that the uncertainties and unfamiliar situations innate to caring for aging parents often evoke fear and anxiety among caregivers. In our study, participants expressed countless unknown factors, including health and legal matters, demonstrating that uncertainties may stem from various sources, including navigating complex healthcare systems, managing chronic health conditions, coping with unpredictable behavioral changes, and addressing financial and legal matters. Having to witness the decline in a loved one's independence and health can exacerbate the fear of the unknown. In addition, caregivers may contend with the fear of making wrong decisions or being

unable to meet their aging parents' evolving needs adequately. These challenges emphasize the importance of providing caregivers with adequate support, resources, and education so they can be empowered to realistically navigate the complexities of caregiving [26,27].

In our findings, navigating the unknown while caring for aging parents involved confronting the many uncertainties and challenges that can arise through the caregiving experience. For instance, the result regarding the unpredictability of health and management of care is similar to that of studies showing that these uncertainties may incorporate various aspects, including the health condition of aging parents, the evolution of their illnesses, availability of suitable resources and support services, and emotional and hands-on demands of caregiving [24,27].

Our results indicate that caregivers often experience increased levels of stress, anxiety, and emotional turmoil because of the uncertainties associated with caregiving. Emotions sometimes cloud a lack of clarity regarding what to anticipate, how to manage evolving health concerns, or how to adapt to changing care requirements, which can evoke feelings of helplessness and being overwhelmed. Similarly, other studies suggest that caregivers frequently struggle with relentless concerns and apprehensions about the well-being of their aging parents, contributing to ongoing stress and potential burnout. The impact of uncertainty in the caregiving role extends across various areas, including emotional, psychological, relational, and physical aspects [26,28].

We highlighted the importance of effective communication within the healthcare system when caring for aging parents, emphasizing its critical role in safeguarding the well-being of both caregivers and older people. Similar studies have emphasized the importance of effective communication, facilitating mutual understanding, encouraging collaboration, and ensuring appropriate care and support. Effective communication is vital for nurturing positive caregiving relationships, facilitating person-centered care, and improving aging parents' quality of life. Caregivers can better deal with the challenges of caring for aging parents with guidance and support when healthcare professionals prioritize clear communication [28,29].

Participants discussed their aging parents' stubbornness and reactions to it, including its effects on care outcomes. Studies show that addressing aging parents' stubbornness presents challenges that warrant patience, empathy, and creative ways of solving problems [30,31]. For example, instances in which an aging parent refuses to adhere to a prescribed medication regimen, despite its importance to health, can be a manifestation of stubbornness. In such situations, caregivers may need to develop strategies involving gentle persuasion. Strategies include involving the parent in the decision-making process and seeking guidance from healthcare professionals to effectively address the issue [30].

Purposeful caregiving involves advocating for the needs and preferences of aging parents in healthcare settings to ensure they receive proper and respectful care [32]. When the participants actively engaged in purposeful caregiving, they aimed to improve their aging parents' overall well-being and quality of life through intentional actions. Studies indicate that purposeful caregiving encompasses a comprehensive strategy, such as demonstrating empathy and empowering and advocating for aging parents [32]. Organizing indoor and outdoor activities for aging parents to promote physical and social stimulation has contributed to their overall well-being [32]. Similar to our study, Mitchell et al. aimed to improve the well-being of aging parents [32].

Meaningful caregiving was described as a blessing [33]. Participants expressed meaningful caregiving as a blessing that gave them a sense of fulfillment and connection. Those who engaged in meaningful caregiving experienced deep gratitude as they supported their aging parents. These participants found deep meaning in their caregiving roles, seizing the opportunity to make a positive difference in their loved ones' lives. For example, caregivers may feel blessed to share intimate moments with aging parents, such as reminiscing about cherished memories, engaging in meaningful activities, or simply providing companionship and emotional support.

Notwithstanding the inherent challenges and sacrifices, the sense of fulfillment from meaningful caregiving often surpassed the difficulties, reinforcing the perception of caregiving as a blessing. Comparably, other studies also identify meaningful caregiving as fostering connection, respect, empowerment, and support in a nurturing and compassionate

manner [34–36]. By espousing these principles and practices, caregivers can create meaningful and fulfilling caregiving experiences for themselves and their aging parents.

## Challenge-Acceptance Theory (CAT) of caregiving

The grounded theory method for this qualitative research generated the challenge-acceptance theory-CAT (Fig 1). To better understand the relationships among the themes used in building the theory, defining the themes relative to the developed theory is imperative. The terminologies and definitions are reflected in Table 3, and the relationships among the themes are depicted in Fig 1.

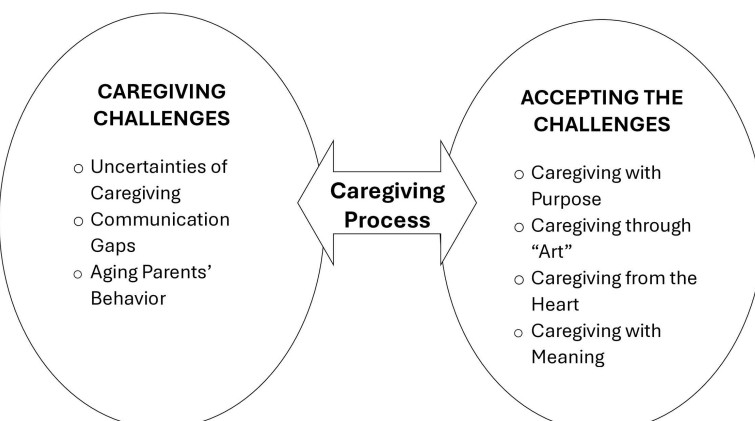

**Fig 1. The Challenge-Acceptance Theory (CAT) of caregiving.**

**Table 3. Terminologies and definitions of themes in the "CAT".**

| Terminology | Definition |
| --- | --- |
| Acceptance | A cognitive and emotional state is achieved by caregivers where they reconcile with the realities and demands of caregiving, allowing them to reframe challenges as meaningful or manageable aspects of their role, thereby reducing the perception of caregiving as a burden. |
| Aging Parents' Behavior | This refers to aging parents' stubbornness, conflicts in personalities and attitudes, and changes in behavior related to the dynamic shifts inherent in the parent-child relationship. |
| The Art of Caregiving (Caregiving through "Art") | Various forms of artistic expression serve as instrumental tools, such as music, visual arts, and dance, to support the unique needs of aging parents and enhance their well-being. |
| Caregiving Process | A series of actions taken and responsibilities upheld by the caregiver to ensure the well-being of their parents aging in place. |
| Challenges | This refers to the difficulties encountered by caregivers during the caregiving process. |
| The Heart of Caregiving (Caregiving from the Heart) | The emotional core of caregiving is that caregivers articulate vital elements such as love, understanding, patience, connection, sacrifice, forgiveness, and preserving dignity and respect as integral components of the caregiving experience. |
| The Need for Effective Communication (Communication Gaps) | Caregivers' need for adequate information and answers to their questions from healthcare professionals regarding their aging parents' health. |
| Purposeful Caregiving (Caregiving with Purpose) | This refers to the intentionality and mindfulness of self-care, that is, caregiving that extends to dedicating time to others and prioritizing personal well-being. |
| The Unknowns (Uncertainties of Caregiving) | This refers to the uncertainties in threading unfamiliar territory and its effect on caregivers. |
| Meaningful Caregiving (Caregiving with Meaning) | This refers to deriving significant meaning from the caregiving experiences, enabling personal growth and fulfillment. |

Using an iterative process of analyzing the data originating from participants' experiences and sharing the findings, the researchers noted a pattern in the case of the participants accepting the caregiving process (Fig 1). Despite the challenges, caregivers learned to embrace and accept the caregiving process with a sense of purpose and found meaning in the role.

The concept of the CAT of caregiving contributes to understanding the experiences of caregivers caring for parents aging in place. The challenge-acceptance relationship is not linear. In this theory, "challenge" refers to the difficulties caregivers face while providing care for their aging parents, but after reaching acceptance, these challenges are no longer perceived as burdensome. Acceptance, therefore, is a cognitive and emotional state achieved by caregivers where they reconcile with the realities and demands of caregiving, allowing them to reframe challenges as meaningful or manageable aspects of their role, thereby reducing the perception of caregiving as a burden.

While challenges may persist throughout the caregiving process, caregivers can navigate in and out of the acceptance phase. In the face of new challenges, caregivers may transition to the caregiving challenge phase. However, they can return to the acceptance phase with adequate support and resources, where the challenges no longer feel burdensome. This dynamic movement between phases is a defining feature of the CAT.

Healthcare professionals can use this theory to assess the caregivers' phase during an encounter. The assessment results will guide healthcare professionals in tailoring their interventions to support caregivers in either the challenge or acceptance phase. For instance, providing resources during the challenge phase can help caregivers transition effectively to the acceptance phase. The CAT also applies to other disciplines that support caregivers of parents aging in place, such as social work and psychology.

The Challenge-Acceptance Theory (CAT) introduced in this study offers a new conceptual framework grounded in the experiences of adult children's caregivers. While elements of CAT echo construct from the stress-coping model [37] and the caregiver burden model [38], it diverges by emphasizing the caregiver's evolving emotional and cognitive journey—from initial distress and uncertainty to the development of coping mechanisms, meaning-making, and eventual acceptance. Unlike previous models that primarily focus on psychological toll and coping strategies, CAT reframes caregiving as a dynamic trajectory rather than a static burden. It integrates emotional distress, cognitive reframing, and the construction of purpose as interconnected stages. This synthesis contributes a unique, process-oriented, and narrative-driven perspective to the caregiving literature, deepening our understanding of how caregivers adapt and find meaning throughout their caregiving experience.

The novelty of CAT lies in its process-oriented and meaning-centered approach. Rather than viewing caregiving solely through the lens of burden or stress management, CAT recognizes the transformative aspects of caregiving that can lead to personal growth and fulfillment. This theory can be practically applied to enhance caregiver support by offering a phase-based framework for assessment, intervention design, and education. For instance, healthcare providers may use CAT to tailor interventions based on a caregiver's current stage, provide anticipatory guidance, or develop training that promotes acceptance-oriented coping strategies. Integrating CAT into caregiver education and counseling may foster resilience, reduce burnout, and improve outcomes for both caregivers and care recipients.

The following two exemplars illustrate how this theory manifests in caregiving experiences and clarify the interrelationships among the themes within the CAT.

### Exemplar 1

Ms. Jones, a 45-year-old female caregiver, cared for her father-in-law, Mr. Smith, an 81-year-old gentleman with dementia. Ms. Jones works as a receptionist at a Walmart store. After caring for her father-in-law for 3 months, working and caring for him was challenging, so she quit her job. Mr. Smith was charming but challenging to care for because he was stubborn (aging parents' behavior). The doctor's visit was also very challenging because Ms. Jones does not have a good knowledge of dementia, symptoms, and sequelae (uncertainties of caregiving). Also, the medical professional did

not communicate the expectations and resources to her, so she relied on what she gleaned from social media, the internet, and her friends (communication gaps). She gets easily frustrated because of Mr. Smith's behavior and continuously watches over him so he does not wander or fall. She could not exercise as she used to and socialize with friends because of Mr. Smith's constant need for care.

With much self-education, she was aware of the resources available to her father-in-law. Ms. Jones followed the lead on these resources and secured four care hours a day for 3 days a week from the Department of Aging to relieve her of the burden of caregiving. With this relief, Ms. Jones started to appreciate the value of the care (caregiving with purpose) she was offering her father-in-law. She had time to think of ways to make him happy, like playing his favorite music and engaging him in artwork (caregiving through "art"). Reviving this interest in Mr. Smith brought a smile to his face, and his behavior toward Ms. Jones changed positively. Because Ms. Jones has realized acceptance to care for Mr. Smith purposefully with love (caregiving from the heart), the challenges of caring for Mr. Smith become bearable.

**Exemplar 2**

Ms. Johnson, a 47-year-old professor, took on the responsibility of caring for her father, Mr. Don, upon his request after her mother passed away from congestive heart failure. Ms. Johnson's parents lived in an independent retirement village, which meant Mr. Don had to relocate not only to be with his daughter but also to a different state. The transition entailed coordinating a plan of care, including locating doctors (facing the unknown). The process involved internet browsing and word-of-mouth (resources) to connect with healthcare professionals. After an arduous process of establishing healthcare personnel to treat Mr. Don's multiple chronic conditions, doctors and nurse practitioners alike find it challenging to decipher why Mr. Don consistently feels fatigued and unwell. Ms. Johnson takes her father to his doctors' appointments and struggles to understand his health conditions. Further lab tests are constantly being recommended.

Mr. Don has some significant health issues because of being in Vietnam back in the 1960s and exposure to Agent Orange. Each day poses a challenge for Ms. Johnson because of the uncertainties of what may happen to Mr. Don, who had fallen several times (fear of the unknown). Ms. Johnson wants her father to have a good quality of life, so she seeks physical therapy for him, enrolls in a senior program, involves friends to check on her father, and endeavors to call her father a couple of times a day while at work (navigating the unknown).

Ms. Johnson feels exhausted and overwhelmed. However, she has a support group (resources) where she shares her feelings and receives encouragement. The support group helped Ms. Johnson look beyond the challenges of caregiving and appreciate the love shared between father and daughter and the wisdom her father imparted to her despite this caring journey (caregiving with meaning).

Three months after the last hospitalization, Mr. Don had another fall and was hospitalized. The fear of having falls was paramount in both Ms. Johnson's and Mr. Don's minds, resulting in Mr. Don's hesitation to drive himself and Ms. Johnson's schedule disruption to care for him. This new caregiving experience resulted in new challenges for the parent and caregiver. It took a month of adjustment in caregiving, physical therapy, and home health care for Ms. Johnson to accept this new phase of Mr. Don's caregiving (caregiving with purpose). Over the years, Ms. Johnson has learned much from her father and observes how he treats others with love and kindness. She feels honored to care for him and is very proud of her father, who sees the best in people. The caregiving process has led to personal growth and increased overall quality of relationships, where Ms. Johnson finds meaning in caregiving (caregiving with meaning).

**Strengths and limitations**

This study explored the experiences of adult children caring for aging parents and developed the Challenge-Acceptance Theory (CAT) to describe the dynamic process caregivers undergo as they transition from initial caregiving challenges to eventual acceptance. The theory provides a conceptual framework to understand how caregivers confront uncertainty, adapt to parental behavioral changes, and navigate emotional and practical demands. Our application of grounded theory

and achievement of thematic saturation reinforces the credibility and trustworthiness of the findings from the participants. The richness and consistency of the data across participants support the robustness of the emergent theory. Therefore, CAT should be considered a meaningful and transferable framework that can inform both future research and caregiving interventions.

Although the sample size limited the possibility of conducting detailed subgroup analyses, no distinct gender differences emerged in the core themes. Given the potential influence of gender norms on caregiving roles and experiences, we recommend that future studies intentionally explore caregiving through a gender-specific lens to uncover nuanced insights into how male and female caregivers may differently navigate the emotional and practical dimensions of caregiving.

Further research is necessary to refine and validate this theory across broader and more diverse caregiving populations. Future studies may expand on this model by including caregivers from different cultural, geographic, and socioeconomic backgrounds to test the generalizability and applicability of the theory. Such work will not only enhance our understanding of caregiving dynamics but also support the development of more targeted interventions to reduce the caregiver burden and promote well-being.

## Conclusion

Our findings emphasize the challenges caregivers face in managing uncertainty and the emotional, physical, and spiritual demands of caregiving. Despite these difficulties, caregivers find meaning in their roles, which leads to a state of acceptance. Their experiences highlight the need for more resources, stronger support systems, and improved communication with healthcare providers. The CAT provides valuable insights into the experiences of adult children caring for their parents who are aging in place and caregivers,in general. The CAT identifies and addresses the specific support needed at each phase of the caregiving journey.

## Acknowledgments

We are grateful to the Catholic University of America professors, the participants who provided insights, and family and friends who supported and encouraged this project.

The authors acknowledge OpenAI's ChatGPT (February 10, 2025 version) for assistance in refining sentence structure and improving readability.

## Author contributions

**Conceptualization:** Oluwakemi Opanubi.

**Formal analysis:** Oluwakemi Opanubi, Jochebed Ade-Oshifogun.

**Methodology:** Oluwakemi Opanubi, Jochebed Ade-Oshifogun.

**Supervision:** Jochebed Ade-Oshifogun.

**Writing – original draft:** Oluwakemi Opanubi, Jochebed Ade-Oshifogun.

**Writing – review & editing:** Oluwakemi Opanubi, Jochebed Ade-Oshifogun.

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
