## [Decision Letter · Decision Letter 0]

7 May 2025

Dear Dr. Opanubi,

Thank you for submitting your manuscript to PLOS ONE. After careful consideration, we feel that it has merit but does not fully meet PLOS ONE’s publication criteria as it currently stands. Therefore, we invite you to submit a revised version of the manuscript that addresses the points raised during the review process.

We look forward to receiving your revised manuscript.

Kind regards,

Maheshkumar Baladaniya

Academic Editor

PLOS ONE

**Journal Requirements:**

1. When submitting your revision, we need you to address these additional requirements. Please ensure that your manuscript meets PLOS ONE's style requirements, including those for file naming. The PLOS ONE style templates can be found at https://journals.plos.org/plosone/s/file?id=wjVg/PLOSOne_formatting_sample_main_body.pdf and https://journals.plos.org/plosone/s/file?id=ba62/PLOSOne_formatting_sample_title_authors_affiliations.pdf 2. Thank you for stating the following in the Acknowledgments Section of your manuscript: We are grateful to the Catholic University of America professors, the participants who provided insights, and family and friends who supported and encouraged this project.The authors acknowledge OpenAI’s ChatGPT (February 10, 2025 version) for assistance in refining sentence structure and improving readability. We note that you have provided funding information that is not currently declared in your Funding Statement. However, funding information should not appear in the Acknowledgments section or other areas of your manuscript. We will only publish funding information present in the Funding Statement section of the online submission form. Please remove any funding-related text from the manuscript and let us know how you would like to update your Funding Statement. Currently, your Funding Statement reads as follows: The author(s) received no specific funding for this work.  Please include your amended statements within your cover letter; we will change the online submission form on your behalf.

**Additional Editor Comments:**

It is good article and should require some minor changes mentioned by the reviewers. Once it is done it is eligible to publish.

Reviewers' comments:

Reviewer's Responses to Questions

**Comments to the Author**

1. Is the manuscript technically sound, and do the data support the conclusions?

Reviewer #1: Yes

Reviewer #2: Yes

Reviewer #3: Yes

2. Has the statistical analysis been performed appropriately and rigorously?

Reviewer #1: N/A

Reviewer #2: Yes

Reviewer #3: N/A

3. Have the authors made all data underlying the findings in their manuscript fully available?

Reviewer #1: Yes

Reviewer #2: Yes

Reviewer #3: Yes

4. Is the manuscript presented in an intelligible fashion and written in standard English?

Reviewer #1: Yes

Reviewer #2: Yes

Reviewer #3: Yes

**Reviewer #1:**  This study is well perceived and executed with a proper methodology. All sections are described properly . However, concluding a research of relatively smaller sample with a theory is too early to make a conclusion. Its true that with out proper training, caring old age people is difficult but it should also be in view that the given example of the study include people over 45 taking care of elderly people. more over this study is specified to middle age people taking care of elderly. what about young people taking care of elderly? different age groups were also to be included to draw a theory as a conclusion

**Reviewer #2: ** Through 22 semi-structured interviews and using Strauss and Corbin’s coding method, the authors developed a novel Challenge-Acceptance Theory (CAT) that explains how caregivers transition from challenge to acceptance. The study presents valuable insights for gerontology, family studies, and health care professionals.

Comments:

The manuscript would benefit from:

A more concise model figure with clearly labeled components.

Clarifying whether this theory is new, how it builds upon or diverges from similar theories (e.g., stress-coping model, caregiver burden model).

Discussing how CAT could be applied practically (e.g., in interventions, assessments, caregiver training).

Consider discussing how the researcher’s background, potential bias, or relationship with participants may have influenced interpretation.

Were bracketing or peer debriefings used to reduce bias?

Discussion:

Compare more explicitly how this study adds new knowledge to existing literature.

Avoid re-stating results in long form—focus on synthesis and implications.

Clarify if any gender-based differences were observed in caregiving experiences (this is implied but not fully explored).

**Reviewer #3: ** - A Data Availability Statement would be formally included to improve openness. Briefly stating that while anonymised extracts of the interview are available and full interview data is secret, additional access may be granted upon reasonable request. A suggested text that you can modify as necessary: "The paper contains the data corroborating the study's findings. Complete interview transcripts are not publicly accessible due to ethical and confidentiality constraints, however they might be provided upon the relevant author's reasonable request, subject to institutional consent.

- Even if it was done informally, think about quickly outlining how you made sure that there was variety in terms of gender and race/ethnicity during snowball sampling (for example, by using targeted outreach or particular recruitment objectives). This will highlight your sampling strategy's resilience even more.

- Kindly indicate which version was utilized (e.g., NVivo 12, NVivo 20).

- As some publications now require it, think about emphasizing the committee name and approval protocol number in the Methods section.

- I suggest adding data about the benefits of physical therapy for improving the health of senior citizens to your discussion of falls and physical exercise. Your discussion of caregiving techniques may be enhanced by research showing the value of mobility and balance training in preventing falls. Studies [DOI: doi.org/10.47363/JPMRS/2023(5)192] that describe how physical therapy therapies might lower fall risks and enhance general health may be of use to you. They offer a useful framework for caregivers of elderly parents who have mobility issues.

**Do you want your identity to be public for this peer review?** For information about this choice, including consent withdrawal, please see our Privacy Policy

Reviewer #1: **Yes: ** Saima Afzal, Ph. D

Reviewer #2: No

Reviewer #3: **Yes: ** Shraddha Baldania

---

## [Author Response · Author response to Decision Letter 1]

6 Jun 2025

Please see the section in the attached document for our response to the reviewers.

---

## [Decision Letter · Decision Letter 1]

29 Aug 2025

Experiences of adult children caring for parents aging in place

PONE-D-25-07300R1

Dear Dr. Opanubi,

We’re pleased to inform you that your manuscript has been judged scientifically suitable for publication and will be formally accepted for publication once it meets all outstanding technical requirements.

Kind regards,

Maheshkumar Baladaniya

Academic Editor

PLOS ONE

Additional Editor Comments (optional):

Manuscript presented good content. It is acceptable for publication.

Reviewers' comments:

Reviewer's Responses to Questions

**Comments to the Author**

Reviewer #2: All comments have been addressed

Reviewer #3: All comments have been addressed

2. Is the manuscript technically sound, and do the data support the conclusions?

Reviewer #2: Yes

Reviewer #3: Yes

3. Has the statistical analysis been performed appropriately and rigorously?

Reviewer #2: Yes

Reviewer #3: Yes

4. Have the authors made all data underlying the findings in their manuscript fully available?

Reviewer #2: Yes

Reviewer #3: Yes

5. Is the manuscript presented in an intelligible fashion and written in standard English?

Reviewer #2: Yes

Reviewer #3: Yes

Reviewer #2: Author had responded to all comments and applied appropriate changes wherever required. It can be accepted.

Reviewer #3: The manuscript's clarity and depth have been much enhanced by the writers' careful consideration of previous reviewer suggestions. The Challenge-Acceptance Theory (CAT) is a significant addition to the literature on caregiving, and the grounded theory methodology is well stated.

**Do you want your identity to be public for this peer review?** For information about this choice, including consent withdrawal, please see our Privacy Policy

Reviewer #2: No

Reviewer #3: No

---

## [Editor Report · Acceptance letter]

PONE-D-25-07300R1

PLOS ONE

Dear Dr. Opanubi,

I'm pleased to inform you that your manuscript has been deemed suitable for publication in PLOS ONE. Congratulations! Your manuscript is now being handed over to our production team.

Kind regards,

on behalf of

Dr. Maheshkumar Baladaniya

Academic Editor

PLOS ONE